# Improved Optical and Electrical Characteristics of GaN-Based Micro-LEDs by Optimized Sidewall Passivation

**DOI:** 10.3390/mi14010010

**Published:** 2022-12-21

**Authors:** Zhifang Zhu, Tao Tao, Bin Liu, Ting Zhi, Yang Chen, Junchi Yu, Di Jiang, Feifan Xu, Yimeng Sang, Yu Yan, Zili Xie, Rong Zhang

**Affiliations:** 1Jiangsu Provincial Key Laboratory of Advanced Photonic and Electronic Materials, School of Electronic Science and Engineering, Nanjing University, Nanjing 210093, China; 2Nanjing National Laboratory of Microstructures, Nanjing University, Nanjing 210093, China; 3College of Electronic and Optical Engineering & College of Flexible Electronics (Future Technology), Nanjing University of Posts and Telecommunications, Nanjing 210023, China; 4Changchun Institute of Optics, Fine Mechanics and Physics, Chinese Academy of Sciences, Changchun 130033, China; 5Xiamen University, Xiamen 361005, China

**Keywords:** Micro-LED, sidewall damage, passivation process

## Abstract

GaN-based Micro-LED has been widely regarded as the most promising candidate for next generation of revolutionary display technology due to its advantages of high efficiency, high brightness and high stability. However, the typical micro-fabrication process would leave a great number of damages on the sidewalls of LED pixels, especially for Micro-LEDs, thus reducing the light emitting efficiency. In this paper, sidewall passivation methods were optimized by using acid-base wet etching and SiO_2_ layer passivation. The optical and electrical characteristics of optimized Micro-LEDs were measured and analyzed. The internal quantum efficiency (IQE) of Micro-LED was increased to 85.4%, and the reverse leakage current was reduced down to 10^−13^ A at −5 V. Optimized sidewall passivation can significantly reduce the non-radiative recombination centers, improving the device performance and supporting the development of high-resolution Micro-LED display.

## 1. Introduction

New display technology using the micron-sized LEDs (Micro-LED or μLED) as pixel units has become the hot topic in the display industry recently. As shown in Figure 1, Micro-LED has higher brightness [1], wider color gamut [2], higher resolution [3,4], greater contrast [5], lower energy consumption [6], longer service life [7], faster response speed [8] and better stability [9] with respect to the Organic LED (OLED). Therefore, Micro-LEDs are considered as the candidate for a wide range of applications, including VR/AR/XR displays of the next generation of display [10,11], visible light communications [12], medical treatment micro devices [13], etc. Definitely, our lifestyle will be changed in the near future.

However, Micro-LEDs with small pixel size encounter certain challenges, such as sidewall damages, luminance efficiency and uniformity [14]. The large body to surface area ratio of Micro-LED will lead to a great sidewall effect, where the etching process causes high density sidewall damages as non-radiative centers. The light emitting efficiency of Micro-LEDs drops significantly in many reports [15,16,17,18]. Therefore, the optimization of sidewall passivation is particularly important for GaN-based Micro-LEDs.

Some methods have been reported to reduce the sidewall etching damages. Seok-In Na et al. used a KOH solution for selective wet etching, which can increase the light output power of LEDs [19]. Sun Y et al. reported that the micro-trenches at bottom corner of mesa could be eliminated by a combined etching treatment using tetramethylammonium hydroxide (TMAH) [20]. Min-Woo et al. found that the leakage current of LED could be reduced by the SiO_2_ passivation layer, where the surface trapping states could be effectively suppressed [21]. S. A. Chevtchenko et al. proved that the combined SiO_2_ and SiN_x_ layers could remove the surface oxygen and surface recombination centers [22]. V. Sheremet et al. also demonstrated that SiO_2_ passivation can reduce the reverse leakage current and increase the light emitting intensity [23]. Kyung Rock Son et al. have demonstrated a 39% higher light output power, 192% higher current density and 63% higher PL efficiency of μLED with optimized SiO_2_ passivation [24]. Chul Huh et al. further utilized (NH_4_)_2_S and (NH_4_)_2_S + t-C_4_H_9_OH to reduce the reverse leakage current [25]. Sabria Benrabah et al. demonstrated that the phosphoric acid solution could repair the surface damages [26]. However, most reported works focused on conventional large-scale LEDs. In addition, the reported light emitting efficiencies of the Micro-LED are still low [27]. Therefore, in order to promote the performance of Micro-LEDs, it is necessary to make a deep investigation of the sidewall passivation method.

In this paper, the passivation method was studied and optimized by combining H_3_PO_4_, KOH and dielectric layers. Optical and electrical characteristics of Micro-LED samples with different passivation methods were measured by photoluminescence (PL), temperature-dependent photoluminescence (TDPL), scanning electron microscope (SEM), current-voltage (I-V) and electroluminescence (EL). Optimized parameters for passivation process were obtained, indicating a high potential in applications including display, visible light communication, and so on.

## 2. Materials and Methods

The commercial blue and green LED used in this work were grown on patterned sapphire substrates by a metal–organic chemical vapor deposition (MOCVD). The structure consisted of 3 μm thick GaN buffer layer, 2 μm thick n-type doped GaN layer, 15 pairs of InGaN/GaN multiple quantum wells (MQW), and a 100 nm thick p-type doped GaN layer. In this experiment, the typical AlGaN electron blocking layer was not applied for the sake of the Micro-LED display application under low injection current.

Firstly, the LED samples were cleaned by ultrasonic cleaning with acetone, ethanol, and deionized water for 10 min, in order to remove the grease and impurities on the surface. Subsequently, 200 nm of the silicon oxide mask layer was grown by plasma-enhanced chemical vapor deposition (PECVD). Micro-sized mesa was patterned by a photolithograph, where reactive ion etching (RIE) and inductively coupled plasma etching (ICP) was used. Then, the buffered oxide etch (BOE) solution was used to remove the residual silicon oxide. In order to investigate the passivation parameters, one Micro-LED sample S1 was kept untreated as the reference sample. The other samples underwent phosphoric acid solution in water baths at the temperature of 25 °C, 50 °C and 80 °C for 10 min, 20 min, 30 min and 60 min, respectively. Finally, five representative samples were prepared with different passivation parameters, as shown in Table 1. Sample S2 was put in phosphoric acid at 80 °C for 60 min; Sample S3 was put in the KOH solution; Sample S4 was placed in a KOH solution and then in the H_3_PO_4_ solution at 80 °C for 60 min; Sample S5 was prepared by an additional SiO_2_ passivation layer grown on the treated sample as sample S4. Finally, the n-type and p-type electrodes were deposited using an electron beam evaporation technique. Then, Ohmic contacts were formed using a typical rapid thermal annealing (RTA) process.

## 3. Results and Discussion

The statistics light emitting intensity of the room temperature PL of samples with different phosphoric acid solution treatments are shown in Figure 2a. Compared to the reference sample without any treatment, the emission intensity of Micro-LED samples can be enhanced 3.5 times by 85% H_3_PO_4_ solution @ 80 °C for 60 min. Figure 2b demonstrates the PL spectra of 10 µm Micro-LED samples S1–S5 with different parameters, as shown in Table 1. It can be observed that sample S5 has the highest luminescence intensity, which is 15-times higher than that of the untreated reference sample S1. We believe the best passivation parameters might be close to that adopted in sample S5. Subsequently the Micro-LED devices were fabricated using such passivation parameters as sample S5, namely the optimized-treated sample. The internal quantum efficiency (IQE) of S1 and S5 samples were tested and shown in Figure 2c,d. The IQE of 10 µm Micro-LED can be enhanced to 85.4%. It proves that the optimized passivation treatment can effectively suppress the non-radiative recombination centers, leading to an enhanced IQE and light emitting intensity.

The time-resolved photoluminescence (TRPL) spectra of sample S1 and S5 were measured using the time-correlated single photon counting (TCSPC), of which the results were shown in Figure 2e. Typically, a double exponential model can be adopted to fit the decay curve. A fast exponential component and a slow exponential component can be acquired [28]:(1)$$I\left(t\right)=A_{1}\;\exp\left(-t/\tau_{1}\right)+A_{2}\;\exp\left(-t/\tau_{2}\right)$$
where *A*_1_ and *τ*_1_ are the fast decay components, and *A*_2_ and *τ*_2_ are the slow decay components. The internal quantum efficiency of LED is highly related to the fast and slow decay components.
(2)$$\eta_{int}^{eq}=\frac{1}{1+\tau_{R}/\tau_{NR}}$$
(3)$$\frac{1}{\tau_{1}}=\frac{1}{\tau_{R}}+\frac{1}{\tau_{NR}}{\;}_{\;}$$
where $\eta_{int}^{eq}$ is the quivalent internal quantum efficiency, $\tau_{R}$ represents the radiative lifetime, and *τ_NR_* represents the non-radiative lifetime [29]. The radiative recombination is sensitive to impurities and defects, and *τ*_1_ is mainly determined by *τ_NR_* as the temperature higher than 40 K. The fast decay component *τ*_1_ of the optimized-treated Micro-LED sample increases from 5.33 ns to 9.88 ns. It indicates that the etching damages and defects on the sidewalls have been removed, thus improving the radiative recombination.

The top-view and cross-sectional SEM images of Micro-LED pixels are shown in Figure 3. It can be observed that the sidewall of the optimized-treated sample S5 is much smoother than that of the untreated sample S1. The focused ion beam technique (FIB) was used to expose the cross-section images of the Micro-LED pixel, as shown in Figure 3c,d. The tilt angle of sidewall is reduced from the original 112° to 93°, which will also reduce the sidewall area. The treatment method used in this paper can modulate the structural characteristics, including the verticality and morphology of the sidewall.

The structural diagram and working images of two fabricated Micro-LED devices were shown in Figure 4a. The effects of the sidewall passivation method on device performance were studied in comparison. As shown, the optimized-treated Micro-LED sample exhibits much higher brightness than that of the untreated sample under the same injected current density. I-V results in Figure 4b prove that the leakage current (from −5 V to 2 V) of the Micro-LED pixel can be reduced by the optimized sidewall passivation process. The dielectric passivation SiO_2_ layer will prevent the contamination of impurities and particles, avoiding the formation of current leakage channels. As shown in Figure 4c,d, the EL intensity of the optimized-treated sample is increased by a factor of 5 times than that of the untreated sample. The EL tests were conducted again after 55 days, and there was a little change in EL intensity. This provides solid evidence that the optimized sidewall passivation could give birth to a high radiation efficiency. It will be useful for high resolution display and high-speed VLC applications.

Figure 5 demonstrates the external quantum efficiency (EQE) of Micro-LEDs with four pixel sizes. It should be mentioned that the improvement in EQE of 10 μm and 20 μm Micro-LED pixels is greater than that of 40 μm and 60 μm pixels due to the large surface to area ratio. Therefore, the sidewall passivation process is vital for the development of Micro-LEDs with small pixel size.

It has been widely reported the ICP dry etching brings a lot of sidewall damages and hanging bonds, resulting in low efficiency. The hanging bonds on sidewall are easily accessible to O^2−^ and OH^−^ ions, forming Ga-O compounds [26]. In this work, KOH solutions can dissolve these unnecessary compounds. The OH^−^ ions have a high kinetic energy at high temperature. Therefore, 80 °C water bath was adopted to increase the passivation rate. This is because the c-plane GaN material is very stable to prevent the wet etching. Thus, the chemical reaction occurs at non-equilibrium positions such as defects, damages and atomic step edges. Next, the phosphoric acid treatment was carried out for further dissolving Ga-O compounds and preventing carbon contamination. It is also helpful for smoothing the sidewall and improving the verticality of the sidewall [30]. Finally, a silicon oxide cover layer was grown to further reduce the probability of the Ga-O bond formation and to prevent contamination by the external environment. As a result, the combined sidewall passivation method can improve the optical and electrical properties of Micro-LED.

## 4. Conclusions

In this paper, a combined H_3_PO_4_ and KOH chemical treatment with the SiO_2_ passivation layer was carried out to reduce the sidewall damages of the Micro-LED caused by the typical micro-fabrication process. The performance of the Micro-LED was significantly enhanced, which was proved by the optical and electrical characteristics of the treated Micro-LED samples. The IQE was enhanced to 85.4%, and the leakage current was reduced. The sidewall passivation treatment could effectively reduce the surface damages and hanging bonds. The results of the EQE further proved that passivation is vital for LED with small pixel size. Optimized passivation method can repair the sidewall damages and suppress the formation of the Ga-O hanging bonds. The improved luminescence efficiency and reliability of Micro-LEDs could support wide applications in display and VLC.

## Figures and Tables

**Figure 1 micromachines-14-00010-f001:**
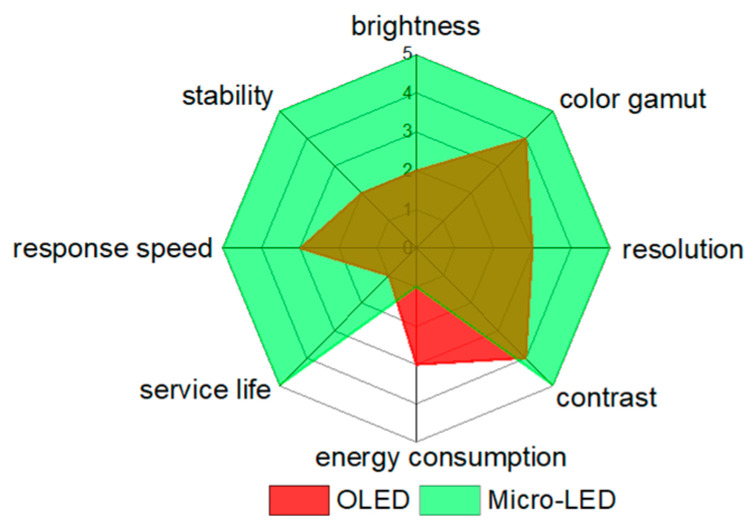
Radar chart of OLED vs. Micro-LED in several aspects.

**Figure 2 micromachines-14-00010-f002:**
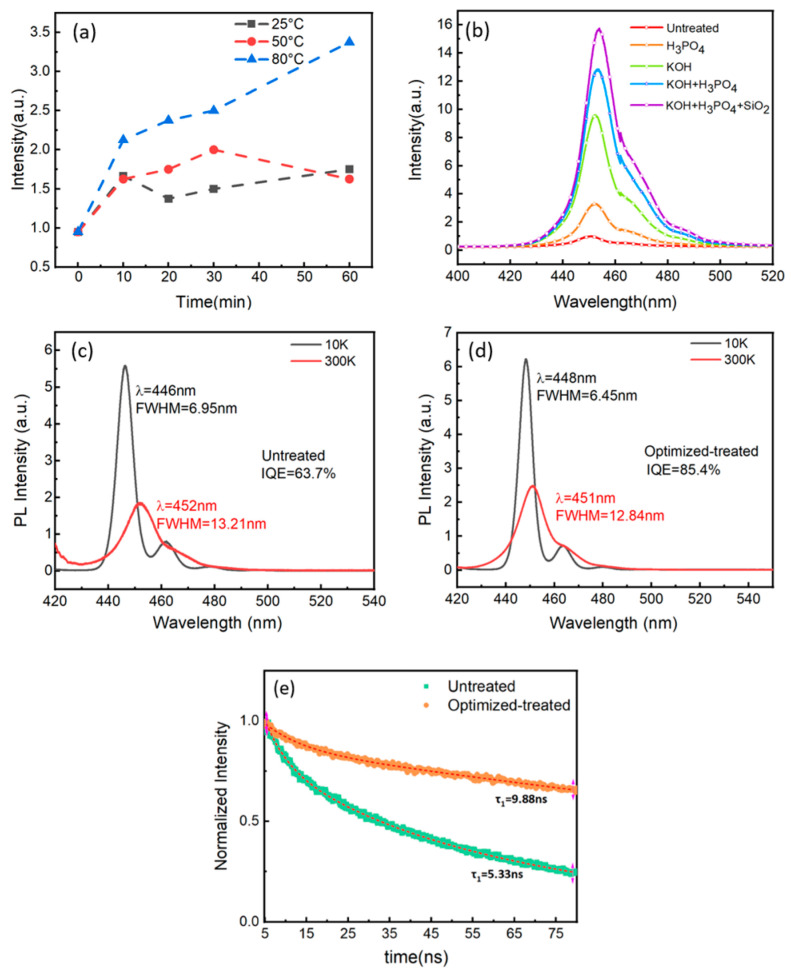
(**a**) Statistics of PL intensity of Micro-LEDs treated by different parameters; (**b**) PL spectra of Micro-LEDs sample S1 to S5; (**c**) TDPL spectra and IQE of untreated sample S1; (**d**) TDPL spectra and IQE of optimized-treated sample S5; (**e**) TRPL of sample S1 and sample S5.

**Figure 3 micromachines-14-00010-f003:**
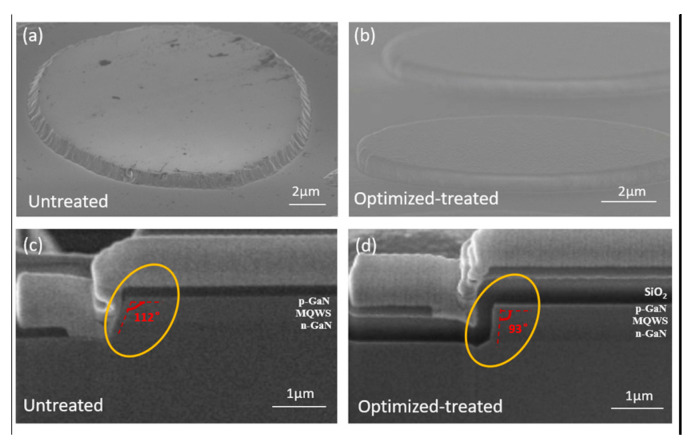
(**a**) SEM images of untreated sample S1; (**b**) SEM image of the optimized-treated sample S5; (**c**) cross-sectional SEM image of sample S1; (**d**) cross-sectional SEM image of sample S5.

**Figure 4 micromachines-14-00010-f004:**
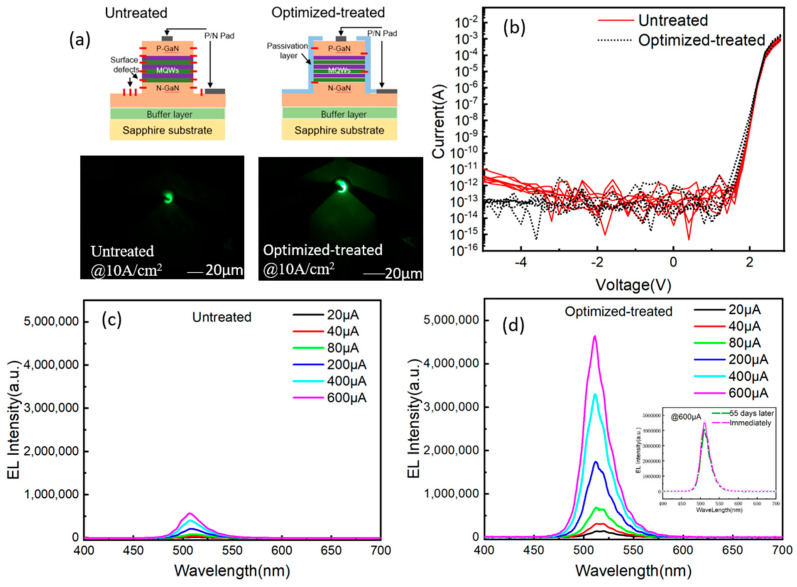
(**a**) Structural diagram and working images of Micro-LED devices; (**b**) Current-Voltage (I-V) curves before and after optimization; (**c**) EL spectra of untreated Micro-LED under the injected current of 600 μA; (**d**) EL spectra of optimized-treated Micro-LED under the injected current of 600 μA.

**Figure 5 micromachines-14-00010-f005:**
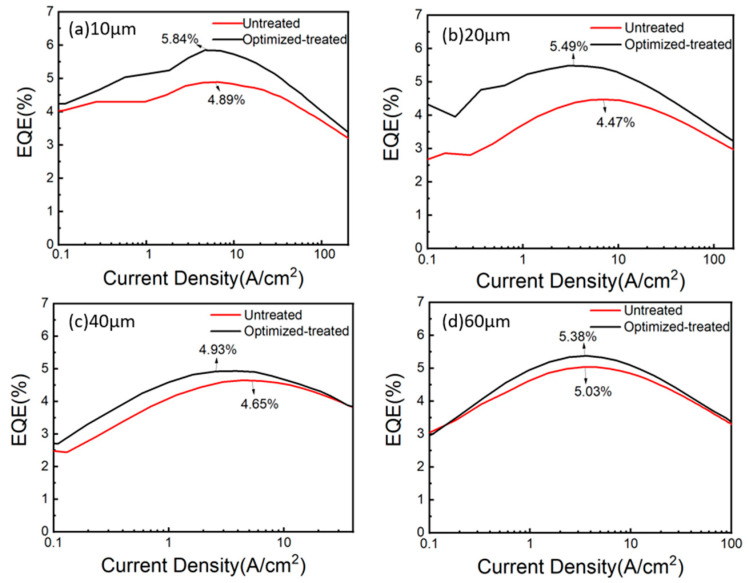
External quantum efficiency for (**a**) the 10 μm Micro-LED pixel; (**b**) the 20 μm Micro-LED pixel; (**c**) the 40 μm Micro-LED pixel; and (**d**) the 60 μm Micro-LED pixel.

**Table 1 micromachines-14-00010-t001:** Optimization parameters of S1–5 samples.

Samples	Parameters of Optimization Methods
S1	Untreated
S2	85% H_3_PO_4_ @ 80 °C 60 min
S3	1 mol/L KOH @ 80 °C 10 min
S4	1 mol/L KOH @ 80 °C 10 min + 85% H_3_PO_4_ @ 80 °C 60 min
S5	1 mol/L KOH @ 80 °C 10 min + 85% H_3_PO_4_ @ 80 °C 60 min + 400 nm SiO_2_ layer

## Data Availability

All relevant data are available from the corresponding author upon reasonable request.

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
