# Peer review of "Improved Optical and Electrical Characteristics of GaN-Based Micro-LEDs by Optimized Sidewall Passivation"

_micromachines, 2022, doi:10.3390/mi14010010_

Round 1

Reviewer 1 Report

The paper is focused on improvement of micro-LEDs by their surface passivation. And authors show that their proposed treatment methods give good results. But I cannot agree with “There is still a lack of suitable sidewall passivation methods applicable to Micro-LED devices”, as this topic is quite popular in recent years papers. And the authors must clear up how this paper is related to their previous one: [Yu, J., Tao, T., Liu, B., Xu, F., Zheng, Y., Wang, X., ... & Zhang, R. (2021). Investigations of sidewall passivation technology on the optical performance for smaller size GaN-based micro-LEDs. Crystals, 11(4), 403].

Also, I have few questions and recommendations:

Do measurements where performed right after the device fabrication? What about the lifetime of the obtained improvement?

Discussion on the physical phenomena that determine improvement of the operation characteristics would greatly improve the scientific significance of the paper.

What happens around -2 V (Fig. 3b), where current drop is observed?

Recommendation: not to start chapter with a figure - text with reference to the figure should go first.

In Fig. 3b and Fig. 4, there are used different curve colors for treated and untreated samples. I suggest to use the same colors over the whole paper.

English needs to be heavily revised.

Author Response

Thank you for your  comments. We have studied comments carefully and have made corrections which we hope meet with approval. Please see the attachment.

Reviewer 2 Report

Authors have proposed a new method of sidewall passivation which enhances the performance of Micro-LED. They also performed optical and electrical characterizations. After optimization, internal quantum efficiency of Micro-LED was enhanced from 63.7 to 85.4% and reverse leakage current was reduced  10-10 A to 10-11 A at -5 V. According to reviewer, this research is systematic and maybe acceptable for the publication in Micromachines. However, there are following problems in this manuscript, which should be addressed before accepting it for the publication in Micromachines.

1.     Please write micro-LEDàMicro-LED in first line of introduction. I would suggest authors to make consistency for the same.

2.     “However, traditional micro-fabrication process, including dry etching, will leave great number of damages on the sidewalls of LED pixels, thus limiting the performance.”  Please change the “will leave” to leaves. Please also check some of the grammatical mistakes in this manuscript.

3.     “Micro-LED display technology using the micron-sized LEDs as light-emitting pixel units, far better than the organic light-emitting diodes (OLEDs) in terms of brightness, resolution, contrast, energy consumption, service life, response speed and stability, has been considered as the candidate for next generation of display[1,2]”. Two references are not sufficient to support this statement. I would suggest authors to add references for each advantage of Micro-LED over OLED. It would be better, if authors provide radar chart/graph for the comparison of Micro-LED and OLED. Please check this reference related issue in rest of the manuscript.

4.     “resulting in great efficiency Droop[7].” Please change Droop to drop.

5.     Statement “The room temperature PL spectra of samples with different phosphoric acid solution treatments were conducted and shown in Figure1a” does not follow the caption of this figure. Both Figure1a and Statement are saying different story. Please also check the caption of Figure 1a. I would suggest author to check the captions of all figures.

6.     The quality of Figure 2 is outstanding. It would be better if authors provide original SEM image which were taken from the instrument. Because these images always have instrument detail, date and time of the experiment, which shows that the images provided in the manuscript are original.

7.     I would suggest authors to increase the font size of X-axes and Y-axes in Figures 3 and 4, which will help readers to get clear picture.

8.     Please provide lifetime curve of treated and untreated Micro-LEDs.  

Author Response

(The authors gave the same response as above.)

Reviewer 3 Report

In this manuscript, the authors have investigated sidewall passivation methods, including acid-base wet etching and SiO2 layer passivation, of which the optical and electrical characterizations were performed and analyzed. They have realized that the internal quantum efficiency (IQE) of Micro-LED is increased from 63.7% before optimization to 85.4%, and the reverse leakage current is reduced from 10-10 A to 10-11 A at -5 V. Also, optimized passivation process can significantly reduce the non-radiative recombination centers on the sidewall, improving the Micro-LEDs performance, and supporting for the development of high-resolution Micro-LED display.

The results achieved in this work are interesting. Therefore, I recommend this paper for publication in this journal after a careful revision.

My comments are as follows:

1.      The language needs to be improved in the manuscript. There are some grammatical as well as typing mistakes. Some sentences need to be rephrased/rewrite. For example, second sentence of abstract and first sentence of introduction. Please avoid too long sentences. I would suggest seeking the help of native English scientific writing expert.

2.      The authors need to add a comparison of this method with the reported methods. I would suggest adding a comparison table that will highlight the effectiveness of this approach with previous ones.

3.      Please avoid using words like “obviously”. In scientific writing, you have to write logically with experimental values and reference. Consider the readers are general.

4.      The authors should add a schematic illustration of device structure.

5.      The authors should make normalized spectra in Fig. 1 and Fig.3.

6.      The results should be explained in more detail. Please provide the reasons of the improvements of results of each characterization. Why this method has shown better results?

Author Response

(The authors gave the same response as above.)

Round 2

Reviewer 1 Report

I would like to thank the authors for their work. They significantly improved the article. I recommend also to include graph with data after 55 days, because for applications this information is important.

Author Response

 Thank you very much for your suggestions! We have included graph with data after 55 days in manuscript.

Reviewer 2 Report

Authors have addressed all my concerns properly. Present form of Manuscript can be accepted for the publication in the micromachines.

Author Response

Thank you for your comments and recognition!